# Fabrication of Manganese Oxide/PTFE Hollow Fiber Membrane and Its Catalytic Degradation of Phenol

**DOI:** 10.3390/ma14133651

**Published:** 2021-06-30

**Authors:** Yan Wang, Diefei Hu, Zhaoxia Zhang, Juming Yao, Jiri Militky, Jakub Wiener, Guocheng Zhu, Guoqing Zhang

**Affiliations:** 1College of Textile Science and Engineering, Zhejiang Sci-Tech University, Hangzhou 310018, China; amywang1021@hotmail.com (Y.W.); hudiefei1027@163.com (D.H.); 2School of Materials Science and Engineering, Zhejiang Sci-Tech University, Hangzhou 310018, China; zhangzx@zstu.edu.cn (Z.Z.); yaoj@zstu.edu.cn (J.Y.); 3School of Materials Science and Chemical Engineering, Ningbo University, Ningbo 315201, China; 4Faculty of Textile Engineering, Technical University of Liberec, 46117 Liberec, Czech Republic; jiri.militky@tul.cz (J.M.); jakub.wiener@tul.cz (J.W.)

**Keywords:** manganese oxide, PTFE hollow fiber membrane, catalyst, degradation, phenol

## Abstract

P-aminophenol is a hazardous environmental pollutant that can remain in water in the natural environment for long periods due to its resistance to microbiological degradation. In order to decompose p-aminophenol in water, manganese oxide/polytetrafluoroethylene (PTFE) hollow fiber membranes were prepared. MnO_2_ and Mn_3_O_4_ were synthesized and stored in PTFE hollow fiber membranes by injecting MnSO_4_·H2O, KMnO_4_, NaOH, and H_2_O_2_ solutions into the pores of the PTFE hollow fiber membrane. The resultant MnO_2_/PTFE and Mn_3_O_4_/PTFE hollow fiber membranes were characterized using scanning electron microscopy (SEM), X-ray photoelectron spectroscopy (XPS), and thermal analysis (TG). The phenol catalytic degradation performance of the hollow fiber membranes was evaluated under various conditions, including flux, oxidant content, and pH. The results showed that a weak acid environment and a decrease in flux were beneficial to the catalytic degradation performance of manganese oxide/PTFE hollow fiber membranes. The catalytic degradation efficiencies of the MnO_2_/PTFE and Mn_3_O_4_/PTFE hollow fiber membranes were 70% and 37% when a certain concentration of potassium monopersulfate (PMS) was added, and the catalytic degradation efficiencies of MnO_2_/PTFE and Mn_3_O_4_/PTFE hollow fiber membranes were 50% and 35% when a certain concentration of H_2_O_2_ was added. Therefore, the manganese oxide/PTFE hollow fiber membranes represent a good solution for the decomposition of p-aminophenol.

## 1. Introduction

Organic wastewater treatment is an important issue owing to its impact on the environment and human health. P-aminophenol, a well-known hazardous environmental pollutant, can remain in water in the natural environment for long periods due to its resistance to microbiological degradation [1,2]. Various symptoms, including skin, eye, and respiratory system irritation, and detrimental effects in the blood and kidneys, were described as a result of p-aminophenol exposure [3].

Conventional methods, such as coagulation, microbial degradation, absorption in activated carbon, incineration, biosorption, filtration, and sedimentation, have been used to treat various kinds of organic wastewater [4,5,6]. Many forms of membrane filter technology, including ultrafiltration, nanofiltration, and reverse osmosis, are commonly used to separate and enrich organic matter from the wastewater, after which the obtained wastes need to be further decomposed into unharmful smaller molecules [7,8,9].

In recent years, a promising approach known as the advanced oxidation process (AOP) was developed to dispose dye wastewater [10]. AOP generally utilizes strong oxidizing species, such as –OH radicals, to trigger a sequence of degradation reactions and break down the dye macromolecule into smaller and unharmful substances [11]. In wastewater treatment, AOPs usually refer to specific subsets of processes that involve O_3_, H_2_O_2,_ and/or ultraviolet (UV) light. However, AOPs can also be used to refer to a more general group of processes that involve photocatalytic oxidation, ultrasonic cavitation, electron-beam irradiation, and Fenton’s reaction [12]. 

In the majority of AOPs, a catalyst is necessary to promote oxidizing species generation. Several metal oxides, such as titanium dioxide (TiO_2_), zinc oxide (ZnO), cerium dioxide (CeO_2_), have been demonstrated to be good catalysts [13,14,15,16]. However, most of the reported catalysis reactions were carried out in homogeneous media, which involves mixing catalysis particles with wastewater, and results in a mixture that is difficult to recycle. 

In order to overcome the disadvantages of the aforementioned methods, fixing the catalytic agent onto a suitable supporting material is a feasible strategy. Organic polymer, zeolites, cellulose fibers, and silica have been reported as suitable supporting materials [16,17,18]. However, there are still several concerns regarding the loading of catalysts onto carriers. Firstly, the catalyst must attach to the carrier tightly and cannot be removed easily, which limits the use of certain highly efficient catalysts, such as TiO_2_ and ZnO. Secondly, it is difficult to encourage the dye molecules in the wastewater to come into contact with catalyst particles, which has a negative effect on catalytic efficiency.

Poly(tetrafluoroethylene) (PTFE) membranes are widely used for water purification [9,19] due to their outstanding chemical stability, high heat resistance, strong hydrophobicity, and fracture toughness [20,21]. The outstanding properties of PTFE are due to the strong C–C and C–F bonds and the carbon backbone, which is protected by a uniform helical sheath formed by the electron cloud of the fluorine atoms [22,23]. Various researchers reported that MnO_2_/CeO_2_ and MnO_2_ exhibit high activity for the complete oxidation of phenol to carbon dioxide and water [24,25,26].

Therefore, in this work, a PTFE hollow fiber membrane (HFM) is proposed as a carrier. Moreover, manganese oxide was selected as the catalyst, which was synthesized in situ in the pores of the PTFE hollow fiber membrane to decompose the p-aminophenol in a consecutive catalytic reaction process. 

## 2. Materials and Methods

### 2.1. Materials

MnSO_4_·H_2_O, NaOH, KMnO_4,_ and 30% H_2_O_2_ at analytical reagent grade were provided by Hangzhou Huipu Chemical Instrument Co., Ltd. (Hangzhou, China), deionized water was purchased from Millipore, p-aminophenol was obtained from Hangzhou Gaojing Chemical Plant, (Hangzhou, China), and PTFE hollow fiber membrane was purchased from Motech Co., Ltd. (Tianjin, China).

### 2.2. Preparations of Manganese Oxide/PTFE Hollow Fiber Membrane

The conditions of the manganese oxide/PTFE hollow fiber membrane preparation were optimized by adjusting the concentration of MnSO_4_·H_2_O, the reaction time, and the temperature. The specific procedures and parameters of the preparation of the MnO_2_/PTFE and Mn_3_O_4_/PTFE hollow fiber membranes are described below.

#### 2.2.1. MnO_2_/PTFE Hollow Fiber Membrane 

The MnSO_4_·H_2_O solution at a concentration of 0.4 mol/L and KMnO_4_ solution at a concentration of 0.1 mol/L were prepared in a digital control ultrasonic cleaner (model: KQ-100DB, supplier: Kunshan ultrasonic instruments Co., LTD, Kunshan, China) at a power of 100 W, a frequency of 40 KHz, and a temperature of 25 °C for 10 min. The MnSO_4_·H_2_O solution was injected into the PTFE hollow fiber membrane using a syringe, and then the PTFE hollow fiber membrane was dried at a temperature of 75 °C for 12 h. After cooling to room temperature, the KMnO_4_ solution was injected into the PTFE hollow fiber membrane using a syringe, and then the hollow fiber membrane was treated by ultrasonic vibration at a power of 100 W, a frequency of 40 KHz, and a temperature of 25 °C for 1 h. This was done so that the KMnO_4_ solution could enter into the micropores of the hollow fiber membrane, and come into adequate contact and produce a complete reaction with MnSO_4_. Thereafter, the PTFE hollow fiber membrane was dried at a temperature of 75 °C for 24 h. Finally, the MnO_2_/PTFE hollow fiber membrane was obtained. The residual MnO_2_ particles in the hollow fiber were washed away using deionized water at a flow rate of 500 mL/h provided by a peristaltic pump, which is shown in Figure 1. 

The reaction mechanism for the synthesis of MnO_2_ is shown as follows: 2KMnO_4_ + 3MnSO_4_ + 2H_2_O = 5MnO_2_ + 2H_2_SO_4_ + K_2_SO_4_(1)

#### 2.2.2. Mn_3_O_4_/PTFE Hollow Fiber Membrane

MnSO_4_·H_2_O solution at a concentration of 0.5 mol/L and NaOH solution at a concentration of 0.05 mol/L were prepared in a digital control ultrasonic cleaner at a power of 100 W, a frequency of 40 KHz, and a temperature of 25 °C for 10 min. The MnSO_4_·H_2_O solution was injected into the PTFE hollow fiber membrane using a syringe, and then the PTFE hollow fiber membrane was dried at a temperature of 75 °C for 12 h. After cooling to room temperature, the NaOH solution was injected into the PTFE hollow fiber membrane using a syringe, and then the PTFE hollow fiber membrane was dried at a temperature of 75 °C for 2 h. After cooling to room temperature, the 30% H_2_O_2_ was injected into the PTFE hollow fiber membrane. The resultant PTFE hollow fiber membrane was placed at a temperature of 75 °C for 24 h. Finally, the Mn_3_O_4_/PTFE hollow fiber membrane was obtained. The residual Mn_3_O_4_ particles in the hollow fiber were washed away using the setup shown in Figure 1. 

The reaction mechanism for the synthesis of Mn_3_O_4_ is shown as follows: MnSO_4_ + 2NaOH = Na_2_SO_4_ + Mn(OH)_2_↓(2)
3Mn(OH)_2_ + H_2_O_2_ = Mn_3_O_4_ + 4H_2_O(3)

### 2.3. Characterization

The cross-section and inner surfaces of the hollow fiber were characterized using field-emission scanning electron microscopy (FE-SEM, Hitachi S-4800, Tokyo, Japan) at 5Kv, and the FE-SEM was equipped with energy dispersive spectroscope (EDS, Oxford instruments, Oxford, UK). The hollow fiber membrane was firstly fractured in liquid nitrogen and then coated with gold using an EMITECH SC7620 sputter coater (Quorum, East Sussex, UK) before SEM testing. The energy dispersive spectrometer (EDS) spectrum (Oxford instruments, Oxford, UK) was recorded simultaneously. The content and distribution of elements were analyzed by EDS and mapping.

The elements, chemical state, and relative content of elements on the sample surface were determined using XPS (Mode: K-Alpha, Company: Thermo Fisher Scientific, Waltham, MA, USA). Before testing, the manganese oxide/PTFE hollow fiber membranes were cleaned with deionized water. 

The thermal degradation processes for PTFE and manganese oxide/PTFE hollow fiber membranes were recorded on a thermogravimetry analyzer (Mode: Pyris 1, Company: Perkinelmer, Waltham, MA, USA) at temperatures from 20–800 °C at a heating rate of 10 °C/min under a 20 mL/min N_2_ gas purge. The temperature of the TGA was periodically calibrated using Al and Fe standards in the range 30–800 °C with an accuracy of ±0.1 °C.

The flux measurement was performed as follows: The flow rate was adjusted using a peristaltic pump to pass the solution through the hollow fiber membrane. The solution exuded out from the membrane wall to obtain the osmotic solution, and the flux was calculated from the mass of the osmotic solution, the surface area of the hollow fiber membrane, and the time. 

The catalytic degradation test was performed as follows: The catalytic performance was estimated using UV spectrum analysis. Concentrations of neutral organic solutes in the feed and permeate solutions were measured using a UV spectrograph. The flux of the hollow fiber membranes was also measured under different operating pressures.

## 3. Results and Discussions 

### 3.1. Morphology of PTFE Hollow Fiber Membrane

The cross-section and inner surface of the PTFE hollow fiber membrane are shown in Figure 2. As can be seen, it is characterized by a multilevel and regular porous structure. The wall thickness of the hollow fiber was about 250 μm. The outer layer is characterized by scarce pores and a large pore diameter of about 5–10 m; the inner layer was smooth and contained dense pores and a small pore diameter of about 1 μm.

### 3.2. Mapping Image of PTFE Hollow Fiber Membrane 

The distribution of manganese oxides in the membrane pores was observed using element distribution analysis. The distribution of C, F, O, and Mn in the cross-section of the manganese oxide/PTFE hollow fiber membrane was obtained using mapping.

As shown in Figure 3, manganese oxide synthesized from MnSO_4_·H_2_O and KMnO_4_ was evenly distributed in the membrane pores. During the preparation of the manganese oxide/PTFE hollow fiber membrane, KMnO_4_ permeated from the inside to the outside, resulting in a higher manganese content in the pores on the inner layer of the membrane.

As shown in Figure 4, manganese oxide synthesized from MnSO_4_·H_2_O, NaOH, and H_2_O_2_ was sparse and stratified in the hollow fiber, and was mainly distributed in the inner and outer layers of the cross-section; the oxygen element was uniform and compact. In the preparation process, the injected H_2_O_2_ permeated from the inside to the outside, so the concentration of H_2_O_2_ in the inner layer was higher, which made it easier to synthesize manganese oxide. Moreover, as a result of the large pore size of the outer membrane, a large amount of oxygen can be stored, which was conducive to the synthesis of manganese oxide.

The results also demonstrate that the content of manganese oxide synthesized in the hollow fiber membrane in the former (Figure 3) was higher than that in the latter (Figure 4).

### 3.3. Electronic Diffraction Spectrum 

KMnO_4_, MnSO_4_·H_2_O and MnSO_4_·H_2_O, NaOH, and H_2_O_2_ were used to synthesize manganese oxide in the PTFE hollow fiber membranes. The element content in the hollow fiber membrane was analyzed using electronic diffraction spectrum (EDS).

According to the EDS spectrum shown in Figure 5, only C, F, Mn, and O were found in the sample, which proved that the manganese element in the mapping came from manganese oxide. The atomic content of each element in the two kinds of manganese oxide/PTFE hollow fiber membranes are shown in Table 1.

The results demonstrate that there were only C and F in the original membrane, and the atomic ratio was about 2:1 (given in Table 1), which is consistent with the atomic ratio of the molecular formula (–CF_2_–CF_2_–)_n_ of PTFE. The atomic percentages of Mn and O in the MnO_2_/PTFE HFM and Mn_3_O_4_/PTFE HFM were approximately 1:2 and 3:4. This reveals that the MnO_2_ and Mn_3_O_4_ were successfully synthesized in the PTFE hollow fiber membranes.

### 3.4. X-ray Photoelectron Spectroscopy 

In order to determine the valence state of Mn in the manganese oxide/PTFE hollow fiber membrane, the manganese oxide/PTFE hollow fiber membranes were analyzed using X-ray photoelectron spectroscopy (XPS), and the oxidation state was determined from the position of the Mn2P binding energy, as shown in Figure 6.

According to Kim’s study [27,28], the Mn2P_3_ (Mn2P_1_) value of the binding energy of Mn_3_O_4_ was 641.01 (653.61), and the Mn2P3 (Mn2P_1_) value of the binding energy of MnO_2_ was 642.61 (654.42). As shown in Table 2, the manganese oxide formed in the PTFE hollow fiber membrane by in situ synthesis was determined as Mn_3_O_4_ and MnO_2_, which is consistent with the experimental results from the simulated hydrothermal method.

### 3.5. TG of PTFE Hollow Fiber with and without Manganese Oxide

A thermal gravimetric analysis of the PTFE, MnO_2_/PTFE, and Mn_3_O_4_/PTFE hollow fiber membranes was carried out. As shown in Figure 7, when the temperature reached 600 °C, it was considered that the base film was completely burned, and the remaining residue represented the content of MnO_2_ and Mn_3_O_4_ in the hollow fiber membrane. As shown in Table 3, more MnO_2_ was loaded in the PTFE hollow fiber membrane than Mn_3_O_4_.

### 3.6. Flux of Membrane

The fluxes of the hollow fiber membranes are shown in Figure 8. The flux of the hollow fiber membranes increased with the increase in the inlet flow rate, and it demonstrated a linear correlation (R^2^ ≥ 0.97 for all hollow fiber membranes).

On the other hand, the fluxes of the PTFE hollow fiber membranes were much higher than that of the others at 7.9, 12.5, 20, 28, and 35 L/m^2^/h corresponding to the flow rates of 175, 200, 230, 265, and 300 mL/h, respectively. The fluxes of the MnO_2_/PTFE hollow fiber membrane were the smallest at 0.43, 1.54, 2, 2.59, and 3.16 L/m^2^/h at the same inlet flow rates. The flux of the Mn_3_O_4_/PTFE hollow fiber membrane was almost twice that of the MnO_2_/PTFE hollow fiber membrane. The reason for the flux decrease could be the occupation of manganese oxide in the pores of the PTFE hollow fiber membrane. Moreover, the content of MnO_2_ was about 25% higher than the content of Mn_3_O_4_ in the PTFE hollow fiber membrane, which can be observed in Figure 7 and Table 3.

On the contrary, the decrease in membrane flux increased the contact time between the manganese oxide and the organic matter and might improve the catalytic degradation efficiency of manganese oxide.

### 3.7. Effect of H_2_O_2_ on the Degradation of Phenol by the Manganese Oxide/PTFE Hollow Fiber Membrane

Oxidants such as hydrogen peroxide (H_2_O_2_) can produce hydroxyl radicals under the catalysis of catalysts. Hydroxyl radicals have a strong oxidation ability, and thus oxidize and degrade phenol.

H_2_O_2_ at concentrations of 0.01 mol/L, 0.02 mol/L, and 0.03 mol/L was added to the phenol solution, which was at a concentration of 50 ppm. The mixed solution was fed into the PTFE, MnO_2_/PTFE, and Mn_3_O_4_/PTFE hollow fiber membranes through the device shown in Figure 1, and the flux was maintained at 1.5 L/m^2^/h. The osmotic solution was collected every 2 h, and tested using a UV spectrophotometer (Thermo Fisher Scientific, Waltham, MA, USA). The results are shown in Figure 9.

It is clear to see that the degradation performance was much higher when manganese oxide was loaded in the PTFE hollow fiber membrane, and the increase in H_2_O_2_ concentration led to a higher degradation rate. When the H_2_O_2_ concentration was 0.03 mol/L, the phenol degradation efficiencies of the MnO_2_/PTFE and Mn_3_O_4_/PTFE hollow fiber membranes were 50% and 35% after 14 h. From the mechanism, it can be inferred that the surface of MnO_2_ was rich in Mn^3+^ and Mn^4+^ [29], while Mn_3_O_4_ was composed of Mn^2+^ and Mn^3+^. This ion distribution produced MnO_2_ and Mn_3_O_4_ in the pores of the PTFE hollow fiber membrane and thus a strong catalytic reaction [30]. This catalyzed H_2_O_2_ to produce a large amount of substances that degrade phenol.

### 3.8. Effect of Peroxymonosulfate on the Degradation of Phenol by the Manganese Oxide/PTFE Hollow Fiber Membrane

According to Edy’s report [31], SO_4_^–^ has a higher redox potential than HO. In the presence of manganese oxide, peroxymonosulfate (PMS) can form a Fenton-like system, which can produce a large amount of SO_4_^–^. SO_4_^–^ has strong oxidability, which has a good degradation effect on phenolic organic compounds [32,33].

PMS at concentrations of 1 × 10^−4^ mol/L, 3 × 10^−4^ mol/L, and 5 × 10^−4^ mol/L was added to the phenol solution, which was at a concentration of 50 ppm. The mixed solution was fed into the PTFE, MnO_2_/PTFE, and Mn_3_O_4_/PTFE hollow fiber membranes through the device shown in Figure 1, and the flux was maintained at 1.5 L/m^2^/h. The osmotic solution was collected every 2 h, and tested using a UV spectrophotometer. The results are shown in Figure 10.

The degradation performance was much higher when manganese oxide was loaded in the PTFE hollow fiber membrane, and the increase in PMS concentration resulted in a higher degradation rate. 

When the PMS concentration was 5 × 10^−4^ mol/L, the phenol degradation efficiencies of the MnO_2_/PTFE and Mn_3_O_4_/PTFE hollow fiber membranes were 70% and 37% after 2 h.

It was hypothesized that the surface of MnO_2_ is rich in Mn^3+^ and Mn^4+^, and Mn_3_O_4_ is composed of Mn^2+^ and Mn^3+^. This ion distribution caused the high MnO_2_ and Mn_3_O_4_ catalytic activity in the catalytic reaction [29], and catalyzed PMS to produce a large number of substances that can effectively degrade phenol [30].

### 3.9. Effect of pH on the Degradation of Phenol by the Manganese Oxide/PTFE Hollow Fiber Membrane

The pH environment has a great influence on the degradation of organic matter by metal oxides [34]. In this experiment, the pH value of the phenol solution with a concentration of 50 ppm was adjusted by combining 1 mol/L sodium hydroxide and 1 mol/L hydrochloric acid. The adjusted phenol solution was fed into the MnO_2_/PTFE and Mn_3_O_4_/PTFE hollow fiber membranes through the device shown in Figure 1, and the flux was maintained at 1.5 L/m^2^/h.

It was found that in the weak acid environment, the degradation of phenol by the two kinds of the manganese oxide/PTFE hollow fiber membrane was higher (as shown in Figure 11). This may be related to the strong capacity of oxygen storage of manganese oxide. Manganese oxide catalyzes oxygen to produce oxygen anion [34,35,36], while H^+^ promotes the degradation of organic matter. In the weak alkali environment, the decrease in phenol was relatively stable. This was hypothesized to be due to the weak acidity of phenol, which can partly react with sodium hydroxide.

### 3.10. Effect of Membrane Flux on the Degradation of Phenol by the Manganese Oxide/PTFE Hollow Fiber Membrane

Membrane flux is not only an important index for characterizing the treatment efficiency of a filtration membrane, but it also affects the catalytic degradation ability of the hollow fiber membrane. According to the following thermodynamic equation: ∆G = ∆H − ∆(S∙T), the chemical reaction takes a certain amount of time. 

The flux of the solution passing through the manganese oxide/PTFE hollow fiber membrane was controlled by the peristaltic pump (as shown in Figure 1). In order to investigate the effect of membrane flux on the phenol degradation efficiencies of the manganese oxide/PTFE hollow fiber membrane, fluxes of 0.5 L/m^2^/h, 1.5 L/m^2^/h, and 3 L/m^2^/h were selected and provided by the peristaltic pump. The phenol solution at a concentration of 50 ppm containing 5 × 10^−4^ mol/L PMS was selected as the inlet solution.

When the membrane flux was 0.5 L/m^2^/h, the phenol degradation rate of the MnO_2_/PTFE and Mn_3_O_4_/PTFE hollow fiber membranes reached 90% and 80%, respectively. When the flux was 1.5 L/m^2^/h, the membrane still demonstrated obvious phenol removal in a short time, but with the increase in time, the phenol degradation effect decreased. However, when the flux was 3 L/m^2^/h, the removal efficiency of phenol was not obvious (as shown in Figure 12). This reveals that it would take a certain reaction time for manganese oxide to catalyze PMS to produce free radicals. The smaller the flux, the more the free radicals that were produced by the manganese oxide to catalyze PMS. A longer reaction time between the organic matter and free radicals would produce an improved reaction.

## 4. Conclusions

In this paper, Mn_3_O_4_/PTFE and MnO_2_/PTFE hollow fiber membranes were successfully prepared by in situ synthesis using two different solution systems under mild conditions and without any surfactants or additives. The flux of the manganese oxide/PTFE hollow fiber membranes decreased as compared with the PTFE hollow fiber membranes; however, their catalytic degradation of phenol improved dramatically. The phenol degradation efficiencies of the MnO_2_/PTFE and Mn_3_O_4_/PTFE hollow fiber membranes were 50% and 35% under the 0.03 mol/L H_2_O_2_ condition, and were 70% and 37% under the 5 × 10^−4^ mol/L PMS condition. In summary, higher H_2_O_2_ concentrations, higher PMS concentrations, weaker acid environments, and smaller fluxes were beneficial for the catalytic degradation of phenol by the manganese oxide/PTFE hollow fiber membrane. Furthermore, this method was demonstrated to be an effective and economic way to decompose p-aminophenol.

## Figures and Tables

**Figure 1 materials-14-03651-f001:**
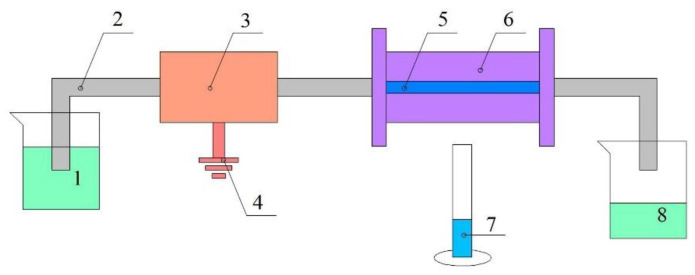
The setup diagram of the manganese oxide/PTFE HFM degradation: (1) p-aminophenol solution or deionized water; (2) silicone tube; (3) peristaltic pump; (4) power supply; (5) MnO_2_/PTFE or Mn_3_O_4_/PTFE HFM; (6) supporter; (7) osmotic solution; (8) p-aminophenol solution or deionized water.

**Figure 2 materials-14-03651-f002:**
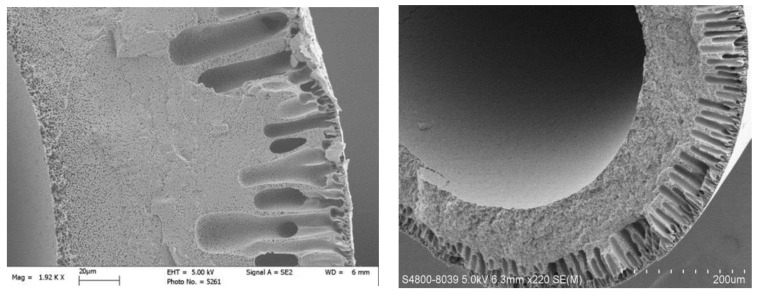
Cross-section and inner surface of the PTFE hollow fiber.

**Figure 3 materials-14-03651-f003:**
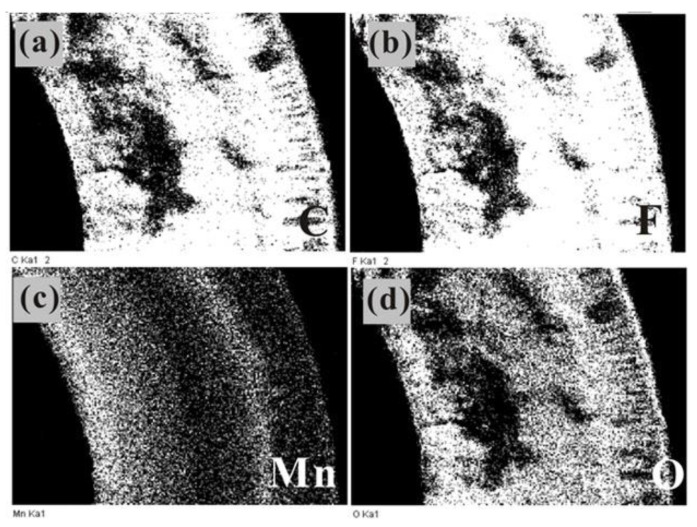
Mapping image of the MnO_2_/PTFE HFM of: (**a**) C element; (**b**) F element; (**c**) Mn element; (**d**) O element.

**Figure 4 materials-14-03651-f004:**
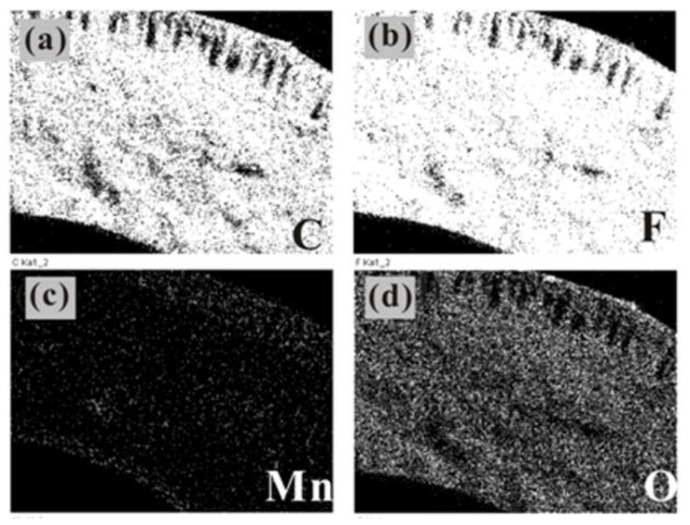
Mapping image of the Mn_3_O_4_/PTFE HFM of: (**a**) C element; (**b**) F element; (**c**) Mn element; (**d**) O element.

**Figure 5 materials-14-03651-f005:**
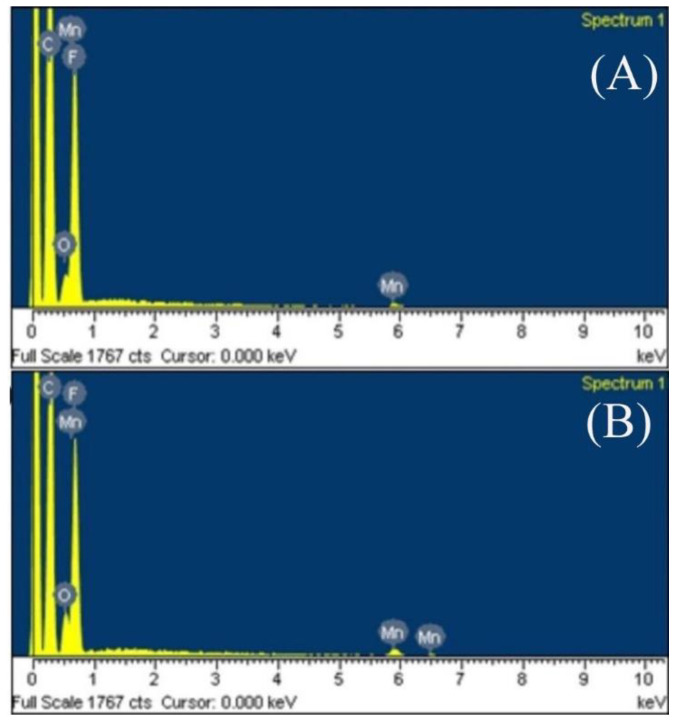
EDS spectrum image of: (**A**) MnO_2_/PTFE HFM; (**B**) Mn_3_O_4_/PTFE HFM.

**Figure 6 materials-14-03651-f006:**
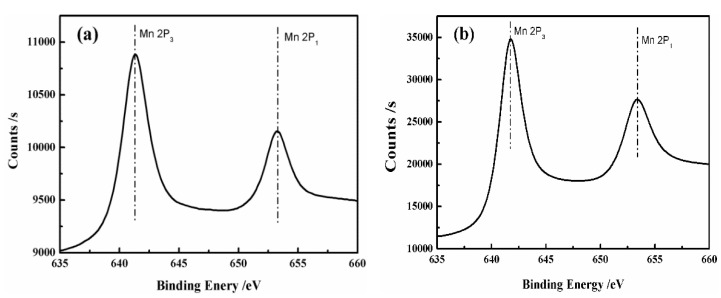
XPS spectra of the Mn2P region of samples: (**a**) Mn_3_O_4_/PTFE hollow fiber membrane; (**b**) MnO_2_/PTFE hollow fiber membrane.

**Figure 7 materials-14-03651-f007:**
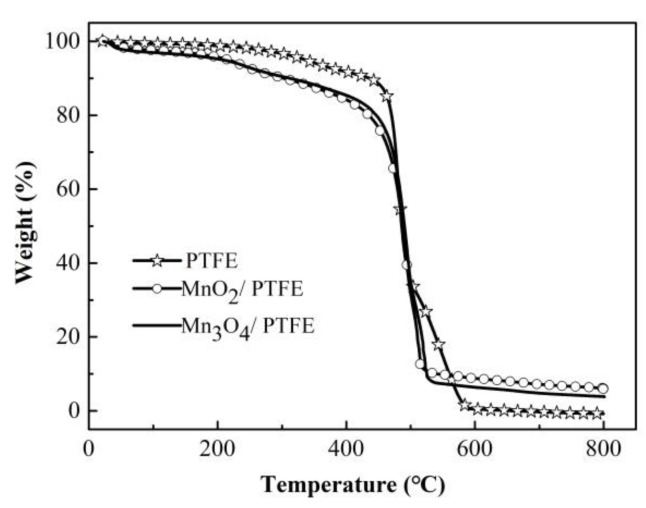
TG analysis of PTFE, MnO_2_/PTFE, and Mn_3_O_4_/PTFE HFMs.

**Figure 8 materials-14-03651-f008:**
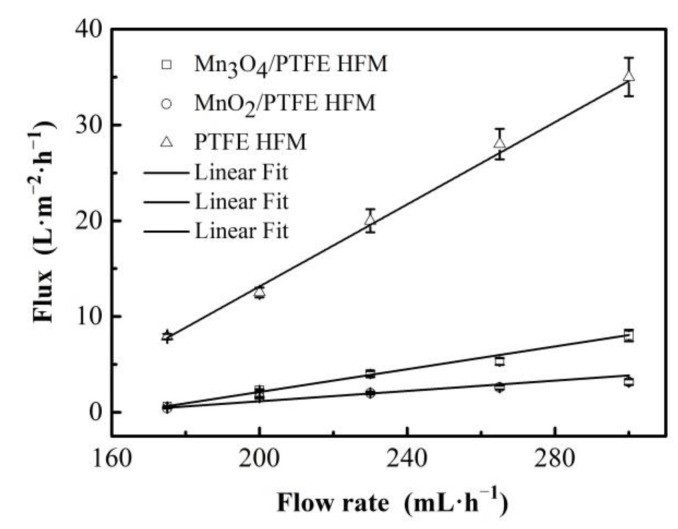
The flux of hollow fiber membranes.

**Figure 9 materials-14-03651-f009:**
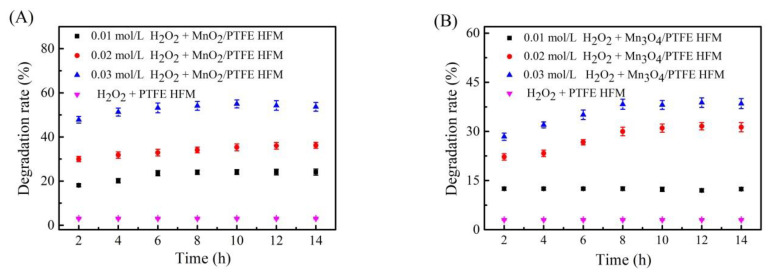
The influence of H_2_O_2_ on the degradation rate of phenol: (**A**) MnO_2_/PTFE HFM; (**B**) Mn_3_O_4_/PTFE HFM.

**Figure 10 materials-14-03651-f010:**
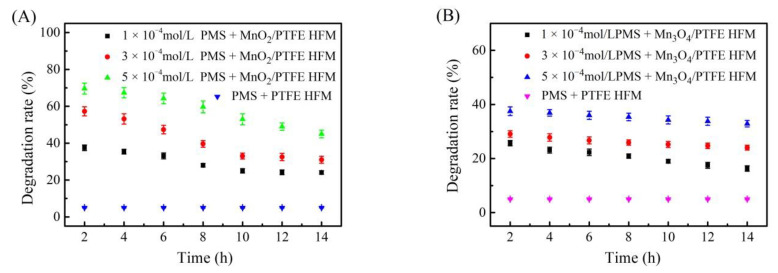
The influence of the concentration of PMS on the degradation of phenol: (**A**) MnO_2_/PTFE HFM; (**B**) Mn_3_O_4_/PTFE HFM.

**Figure 11 materials-14-03651-f011:**
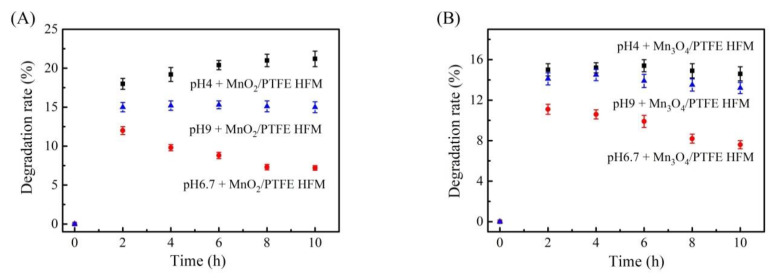
The influence of pH on the degradation of phenol: (**A**) MnO_2_/PTFE HFM (**B**) Mn_3_O_4_/PTFE HFM.

**Figure 12 materials-14-03651-f012:**
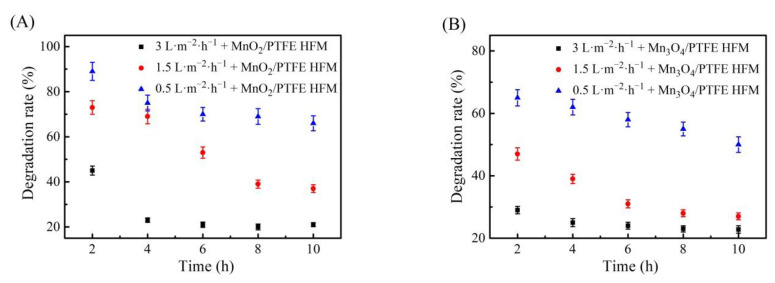
The influence of flux on the degradation of phenol: (**A**) used MnO_2_/PTFE HFM; (**B**) Mn_3_O_4_/PTFE HFM.

**Table 1 materials-14-03651-t001:** The element contents of MnO_2_/Mn_3_O_4_ PTFE HFMs.

Samples	Element Content/%			
	C	F	O	Mn
PTFE HFM	68.61	31.39	/	/
MnO_2_/PTFE HFM	58.92	32.19	5.73	3.16
Mn_3_O_4_/PTFE HFM	59.08	30.78	5.64	4.5

**Table 2 materials-14-03651-t002:** The binding energy of Mn 2P of MnO_2_ and Mn_3_O_4_.

Samples	MnO_2_	Mn_3_O_4_
Mn 2P_3_(eV)	642.6	641.0
Mn 2P_1_(eV)	654.4	653.6

**Table 3 materials-14-03651-t003:** Manganese oxide content in the PTFE HFMs.

Samples	MnO_2_/PTFE HFM	Mn_3_O_4_/PTFE HFM	PTFE HFM
Residual content at 600 °C (%)	8.78	6.47	0.28

## Data Availability

Data available in a publicly accessible repository.

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
