# Peer review of "Fabrication of Manganese Oxide/PTFE Hollow Fiber Membrane and Its Catalytic Degradation of Phenol"

_materials, 2021, doi:10.3390/ma14133651_

Round 1
Reviewer 1 Report
Manuscript Number: materials-1248115
Manuscript title: Fabrication of manganese oxide / PTFE hollow fiber membrane 2 and its catalytic degradation of phenol
Journals: Materials (ISSN 1996-1944)
The manuscript submitted by authors are interesting and within the scope of Journal. But manuscript lack in several ways.
- Abstract section need to rewrite again. While writing the abstract just follow the important instruction. An abstract summarizes, usually in one paragraph of 300 words or less, the major aspects of the entire paper in a prescribed sequence that includes: 1) the overall purpose of the study and the research problem(s) you investigated; 2) the basic design of the study; 3) major findings or trends found as a result of your analysis; and, 4) a brief summary of your interpretations and conclusions.
- Provide the clear statement under introduction section describing the one or more key hypotheses that the work described in the manuscript was intended to confirm or refute. Inclusion of a hypothesis statement makes it simple to contrast the hypothesis with the most relevant previous literature and point out what the authors feel is distinct about the current hypothesis (novelty).
- Introduction section need to be revised. I can find most of the sentences are without references. Kindly provide the appropriate references.
- On page 1, author mention “Organic wastewater treatment has been a concerned issue owing to its impact on environmental and health problems. Conventional methods such as coagulation, microbial degradation, and absorption on activated carbon, incineration, biosorption, filtration and sedimentation have been used to treat various organic wastewater”. Provide the reference. [Science of The Total Environment, 147851, 2021].
- On page 3,” In wastewater treatment, AOPs usually refer to specific subsets of processes that involve O3, H2O2 and/or UV light. However, AOPs could also be used to refer to a more general group of processes that involve photocatalytic oxidation, ultrasonic cavitation, electron-beam irradiation and Fenton’s reaction. Provide references”. Photocatalysts in Advanced Oxidation Processes for Wastewater Treatment, John Wiley & Sons (USA) 1, 320
- Some more detail information and literature survey related to PTFE hollow fiber membrane need to be added.
- On page 2, section 2.2 Preparations of manganese oxide / PTFE hollow fiber membrane, it author own protocol or adopted from literature. If its author protocol discuss the optimization for Preparations of manganese oxide / PTFE hollow fiber membrane. If not, provided the appropriate reference.
- On page 3, section 2.3 Characterization, provide the made and operating conditions of all the instruments used in this work.
- Provide all the graphs with error bar.
- A schematic mechanism for the synthesis and application in phenol degradation need to be added and discuss in details.
- Provide the regeneration study?
- How practical and cost effective in real life application.
Author Response
Dear Reviewer,
Thanks a lot for your valuable comment. According to your comments, I have modified my manuscript. my response to your comments is attached.
- Agreed and modified. The revised content is marked in blue in abstract section.
-
Agreed and modified in the introduction section. The novelties are: (1) the synthesis and storage of manganese oxide in hollow fiber membrane; (2) the consecutive catalytic reaction provided by the manganese oxide/PTFE hollow fiber membrane.
-
Agreed and modified through the whole manuscript.
-
Agreed and added.
-
Agreed and added.
-
Agreed and added in the introduction section.
-
The information is added in the section 2.2.
It’s author’s own protocol.
The conditions of manganese oxide / PTFE hollow fiber membrane preparation were optimized by adjusting the concentration of MnSO4·H2O, the reaction time, and temperature.
-
Agreed and modified.
-
Agreed and added.
-
Agreed and added.
2KMnO4 + 3MnSO4 + 2H2O == 5MnO2 + 2H2SO4 + K2SO4
(1)
MnSO4 + 2NaOH = Na2SO4 + Mn(OH)2↓
(2)
3Mn(OH)2 + H2O2 = Mn3O4 + 4H2O
(3)
- We haven’t done this yet. We will continue this work in near future.
-
Based on our experimental results, it’s practical and economic.

Reviewer 2 Report
In this contribution, Wang et al. presented a comprehensive work on the preparation manganese oxide / PTFE hollow fiber membrane. I am in favor of publishing this manuscript containing interesting data. My comments are given below for a major revision:
- “Therefore, in this work the hollow fiber membrane was proposed to be a carrier, and manganese oxide was selected as the catalyst which…” – Authors did not clarify the reason of selecting manganese oxide for this study. No base line information/literature survey was provided in the Introduction section. Please include.
- A clear statement on the novelty of the research is missing. Please state what is new in this study.
- “The MnSO4·H2O solution at a concentration of 0.4 mol/L and KMnO4 solution at a concentration of 0.1 mol/L were prepared at room temperature by ultrasonic dissolving method.” – Please mention the sonication treatment time and provide with some details on the sonicator (manufacturer, power used, etc.).
- “The MnSO4·H2O solution was injected into the PTFE hollow fiber membrane by using a syringe, and then the PTFE hollow fiber membrane was dried at a temperature of 75℃ for 12 hours.” – Did the authors optimize the time and temperature? What would happen if both are increased or decreased?
- “… the hollow fiber membrane was treated by ultrasonic vibration for 1 hour” – Please discuss the reason of this treatment in the text. Provide some details on the sonicator if this instrument is different than that of the used in the earlier treatment.
- “And the residual MnO2 particles in the hollow fiber were washed away by a setup shown in Figure 1.” – This step is not clear from the diagram. Please discuss in details.
- Why did the authors use 12 hours drying time for MnO2/PTFE hollow fiber but 24 hours drying time for Mn3O4/PTFE hollow fiber?
- “At last, the MnO2/PTFE hollow fiber membrane was obtained.” and “At last, the Mn3O4/PTFE hollow fiber membrane was obtained.” – These are not clear at all. What is the chemistry behind these formation? Please show all reactions involved and explain.
- Why did the authors dry the PTFE hollow fiber membrane and then performed the washing of residual MnO2 particles?
- “…manganese element was sparse and stratified in the hollow fiber, mainly distributed in the inner and outer layers of the cross section…” – How was this element distributed “mainly” in the inner and outer layer? How can this distribution be controlled? Please explain the mechanism.
- Fig. 7: It would not be possible to differentiate the lines if printed black and white. Please revise.
- Section 3.5: “… the flux of Mn3O4/PTFE hollow fiber membrane was almost two times higher than that of the MnO2/PTFE hollow fiber membrane.” – Why?
- The English language of this manuscript needs improvement. Some description and explanations are not clear throughout the manuscript which can be improved. I am recommending to use a scientific language editing service.
Author Response
Dear Reviewer,
Thanks a lot for your valuable comment. According to your comments, I have modified my manuscript. My response to your comments is attached.

Reviewer 3 Report
Dear Authors,
The results discused in this paper represent a real interest for the decontamination of wastewater by using the hollow fiber membrane proposed to be a carrier and manganese oxide as a catalyst which has been synthesized in situ in the pores of the hollow fiber membrane of PTFE.
True, the catalyst must be tightly attached to the carrier and cannot be easily removed. Second, the dye molecules in the wastewater must be in good contact with the catalyst particles, in order to have a positive effect on the catalytic efficiency.
The question would be, has it been tested how long or after how many cycles of use manganese oxide remains in the pores of the hollow fiber membrane of PTFE?
there are no tests and comments to prove that these MnO2/PTFE 274 and Mn3O4/PTFE hollow fiber membranes, after how many cycles / hours / days of operation are still effective. In other words, how efficient the catalyst remains after a certain period of operation.
At least a mapping of the chemical elements with EDX must be redone, after a period to be able to observe the presence of the catalyst.
Author Response
Dear Reviewer,
Thanks a lot for your valuable comments. According to your comments, I have modified my manuscript. My response to your comments are as follows:
The samples have been tested for 14 hours. And after 14 hours, they were still effective. The conclusion can be proved from figure 9-12, there were very small decrease in catalytic degradation rate as the time passing.
Round 2
Reviewer 1 Report
The authors have revised the script. But still two of the (comments 11 and 12) were not work in revising the script. These two comments are really very important to improve the overall quality of manuscript.
- Provide the regeneration study?
- How practical and cost effective in real life application.
Author have responded for comment 12, “Based on our experimental results, it’s practical and economic”. I do not understand how author proof based on the experiment that this material is practrical and economic. To confirm this, author have to do the experiment, which are proposed in comments 11. Comment 12 answer, depend upon comment 11 study. Apart from that industrial real wastewater are needed to perform the catalytic degradation study using manganese oxide / PTFE hollow fiber membrane.
Author Response
Dear Reviewer,
Thanks a lot for your valuable comment. My responses to your comments 11 and 12 are:
Comment 11: Provide the regeneration study?
Response to 11: The current work was aim to investigate the possibility of synthesis of manganese oxide in PTFE hollow fiber membrane, and to construct a continual filtration and catalytic degradation process for wastewater treatment. And we will continue the regeneration study in near future.
Comment 12. How practical and cost effective in real life application.
Response to 12: In this work, the traditional and commercial hollow fiber membrane was functionalized with manganese oxide. Therefore, the hollow fiber membrane had the functions of filtration and catalytic degradation. The MnSO4·H2O, NaOH, KMnO4, and H2O2 are common materials, and the process for manganese oxide synthesis is simple, therefore, the preparation of manganese oxide/PTFE hollow fiber membrane is economic. Besides, this catalytic degradation is continual. Therefore, it could be used for industrial application. However, as the reviewer mentioned, we still need more work to go further, such as regeneration study, mass production, application in industrial conditions.
Besides, the manuscript has been edited by english editing services, and the certificate is attached.

Reviewer 2 Report
Acceptance is recommended.
Author Response
Dear reviewer, thanks a lot for accepting my paper.
The manuscript has been edited by an English editing service, the certificate is attached.

Reviewer 3 Report
Dear Author,
After your changes, I consider that the paper can be published.
Author Response

(The authors gave the same response as above.)

Round 3
Reviewer 1 Report
The authors have satisfactorily revised the script.
Author Response
Dear reviewer,
Thanks a lot for your valuable comments and acceptance.